# From Infection to Infertility: Diagnostic, Therapeutic, and Molecular Perspectives on Postpartum Metritis and Endometritis in Dairy Cows

**DOI:** 10.3390/ani15192841

**Published:** 2025-09-29

**Authors:** Ramanathan Kasimanickam, Priunka Bhowmik, John Kastelic, Joao Ferreira, Vanmathy Kasimanickam

**Affiliations:** 1College of Veterinary Medicine, Washington State University, Pullman, WA 99164, USA; vkasiman@wsu.edu; 2College of Agricultural, Human, and Natural Resource Sciences, Pullman, WA 99163, USA; priunka.bhowmik@wsu.edu; 3Faculty of Veterinary Medicine, University of Calgary, Calgary, AB T2N 4Z6, Canada; jpkastel@ucalgary.ca; 4School of Veterinary Medicine and Animal Science, São Paulo State University-UNESP, Botucatu 18618-81, SP, Brazil; joao.cp.ferreira@unesp.br

**Keywords:** dairy cows, uterus, metritis, systems biology, biomarkers, fertility

## Abstract

**Simple Summary:**

After calving, many dairy cows develop uterine infections like metritis and endometritis which can lower fertility and cause financial losses. They are caused by harmful bacteria and a complex immune response, complicating diagnosis and treatment. Inflammation also damages the uterus, delaying re-establishment of pregnancy. Antibiotics like ceftiofur are commonly used, but their efficacy is decreasing due to antibiotic resistance and formation of protective bacterial layers (biofilms). New research tools, i.e., proteomics, metabolomics, and microRNA studies, document changes in immune proteins, energy use, and inflammation, providing new knowledge regarding how these diseases affect the cow. Applying this knowledge, along with better diagnostics and supportive treatments (vitamins and immune boosters), will improve how we manage these diseases. This should lead to healthier cows, better fertility, and fewer losses for the dairy industry.

**Abstract:**

Postpartum uterine diseases such as metritis and endometritis impair reproductive performance and cause substantial economic losses in dairy cows worldwide. The multifactorial etiology, involving polymicrobial infections and complex host immune responses, poses diagnostic and therapeutic challenges. Traditional treatments rely on antibiotics, e.g., cephalosporins like ceftiofur and cephapirin, with broad-spectrum efficacy. However, emerging antimicrobial resistance, biofilm formation by pathogens such as *Trueperella pyogenes*, *Fusobacterium necrophorum*, and *Escherichia coli*, and bacterial virulence factors have reduced effectiveness of conventional therapies. Advances in systems biology, particularly proteomics, metabolomics, and microRNA (miRNA) profiling, have provided unprecedented insights into the molecular mechanisms underpinning uterine disease pathophysiology. Proteomic analyses reveal dynamic changes in inflammatory proteins and immune pathways, whereas metabolomics highlight shifts in energy metabolism and bacterial–host interactions. Furthermore, miRNAs have critical roles in post-transcriptional gene regulation affecting immune modulation, inflammation, and tissue repair, and also in modulating neutrophil function and inflammatory signaling. Uterine inflammation not only disrupts local tissue homeostasis but also compromises early embryo development by altering endometrial receptivity, cytokine milieu, and oocyte quality. Integration of multi-omics approaches, combined with improved diagnostics and adjunct therapies—including micronutrient supplementation and immunomodulators—offers promising avenues for enhancing disease management and fertility in dairy herds. This review synthesizes current knowledge on proteomics, metabolomics, and miRNAs in postpartum uterine diseases and highlights future directions for research and clinical applications.

## 1. Introduction

Postpartum uterine diseases remain a critical challenge in dairy production systems worldwide, exerting profound negative impacts on animal health, reproductive performance, and farm profitability. After calving, the bovine uterus undergoes a complex involution process, during which it is particularly susceptible to bacterial contamination and inflammation [1,2]. Common postpartum uterine disorders, such as metritis and endometritis, occur when uterine involution is disrupted by infectious agents and an exaggerated or unresolved inflammatory response [3]. These conditions are among the leading causes of infertility in dairy cows, manifesting as delayed return to estrus, reduced conception rates, prolonged calving intervals, and increased culling risk [4]. Consequently, postpartum uterine diseases impose big economic losses on the dairy industry by diminishing reproductive efficiency and milk yield [5].

The etiology of uterine disease in postpartum cows is multifactorial and dynamic. It involves an intricate interplay among opportunistic pathogens, innate and adaptive immune defenses, environmental stressors, and nutritional status [6]. Pathogens such as *Trueperella pyogenes*, *Fusobacterium necrophorum*, and *Escherichia coli* frequently dominate uterine infections and are often present in polymicrobial communities that can form biofilms [7,8,9,10], enhancing bacterial persistence and resistance to antimicrobial therapy, complicating disease resolution [9,10]. Additionally, host inflammatory response, necessary for bacterial clearance, can damage tissue and delay uterine healing if dysregulated [11,12]. This delicate balance between infection control and immune-mediated tissue repair underscores the complexity of postpartum uterine diseases and challenges in predicting and managing their progression [11,12].

Diagnosis and treatment of postpartum uterine diseases continue to pose considerable challenges. Traditional diagnostic methods, e.g., clinical examination, ultrasonography, and cytology, can be subjective and often fail to detect subclinical infections or predict disease severity [13]. Therapeutic strategies primarily rely on antibiotic administration, including systemic or intrauterine cephalosporins, efficacious against many uterine pathogens [14]. However, increasing antimicrobial resistance and biofilms highlight limitations of current treatments. Furthermore, indiscriminate antibiotic use raises concerns about drug residues in milk and implications for antimicrobial stewardship [15,16].

Advances in high-throughput molecular technologies have revolutionized understanding of postpartum uterine diseases. Systems biology approaches, including proteomics [17,18] metabolomics [19,20] and microRNA (miRNA) profiling [21,22,23], offer comprehensive insights into the molecular and biochemical alterations associated with uterine health and disease. Proteomic analyses confirmed dynamic changes in protein expression related to immune responses, inflammation, and tissue remodeling during the postpartum period [17,18]. Metabolomic profiling of blood, uterine fluids and milk revealed shifts in energy metabolism and inflammatory mediators that reflect host–pathogen interactions and uterine microenvironment changes [19,20]. Additionally, miRNAs—small, non-coding RNA molecules that regulate gene expression post-transcriptionally—are key modulators of immune and inflammatory pathways, with several miRNAs identified as potential biomarkers and therapeutic targets in uterine disease [21,22,23].

Beyond uterine health, postpartum infections also have profound effects on early embryonic development and fertility. The uterus provides a specialized environment essential for embryo survival, growth, and maternal recognition of pregnancy. Inflammatory insults and altered metabolic and immune signaling within the uterine milieu can impair endometrial receptivity and embryo viability, increasing embryonic loss, even after clinical resolution of uterine infection [24,25]. This intersection of uterine disease with reproductive success highlights importance of timely diagnosis and effective management.

Despite progress, postpartum uterine diseases remain difficult to predict, diagnose, and manage. Traditional diagnostic and therapeutic approaches are limited by subjectivity, resistance, and sustainability concerns. However, molecular insights from proteomics, metabolomics, and miRNA profiling provide unprecedented opportunities to refine disease detection, prognosis, and treatment.

This review synthesizes current knowledge on postpartum uterine diseases in dairy cows, with an emphasis on integrating molecular evidence with clinical perspectives. By highlighting recent advances in systems biology and their translational potential, we seek to advance both scientific understanding and practical strategies for managing uterine health, enhancing reproductive performance, and improving sustainability in the global dairy industry.

## 2. Postpartum Uterine Disease

Postpartum dairy cows have a risk of uterine infection due to anatomical barriers breached during and after calving [8,26]. Parturition disrupts reproductive tract integrity, making it more susceptible to pathogen entry [8,27,28,29]. However, the cow’s immune system works to eliminate these pathogens both locally and systemically, with local responses typically focused on the uterus [30,31]. Regardless, in some cows, pathogens are not fully cleared from the uterus; these persistent uterine infections adversely affect cow health, reproductive performance, and productivity [32]. If these infections are not adequately managed, they can prolong recovery, decrease fertility, and cause health issues that compromise cow well-being and efficiency of the dairy operation.

### 2.1. Risk Factors

Several risk factors can increase the likelihood of uterine infections in postpartum dairy cows, largely due to physical and metabolic stress during and after calving [33,34]. Dystocia is a major risk factor as it often leads to uterine trauma, predisposing to infection. Similarly, cows birthing twins are at an increased risk of various postpartum issues, particularly retained fetal membranes, often followed by uterine infections.

Other metabolic disorders such as periparturient hypocalcemia, ketosis, and displaced abomasum further elevate the risk of uterine infections [35,36]. Periparturient hypocalcemia (low blood calcium concentrations) weakens the immune system and impairs its ability to respond to uterine infections. Ketosis, elevated blood ketone concentrations due to negative energy balance, also suppresses the immune response, making cows more susceptible to infections, including in the uterus. Displaced abomasum can decrease feed intake and cause metabolic imbalance, compromising postpartum recovery. When these risk factors are present, uterine infections are more likely, with potential to negatively impact reproductive health, milk production, and overall well-being.

### 2.2. Pathogens

Uterine infections in postpartum dairy cows are linked to various pathogens, each contributing to various uterine diseases [37,38,39,40]. *Trueperella pyogenes* (formerly *Arcanobacterium pyogenes*) is frequently associated with metritis and endometritis and can cause persistent inflammation. *Escherichia coli* is commonly present in uterine infections, particularly after calving. It can lead to acute infections with rapid onset and systemic illness, e.g., fever or lethargy. *Prevotella melaninogenicus* is a Gram-negative bacterium typically involved in mixed infections of the uterus. It thrives in anaerobic environments and often accompanies other bacteria, contributing to persistent uterine infections. Similarly, *Fusobacterium necrophorum*, another anaerobic pathogen, causes necrotic lesions in the uterus and is known to cause severe inflammation, particularly with retained fetal membranes or after dystocia. Bovine Herpesvirus 4 (BoHV-4) can impair the immune response, making the uterus more susceptible to secondary bacterial infections [41]. BoHV-4 is often associated with inflammation of the reproductive tract and can cause viral endometritis. These pathogens highlight the complex nature of uterine infections in dairy cows, with bacterial and viral agents contributing to the overall risk of reproductive failure and health complications in affected animals.

### 2.3. Pathophysiological Mechanisms

The underlying mechanism by which pathogenic bacteria in the uterus cause uterine infections involves a cascade that negatively impacts reproductive function [37,38,39,40,41]. When pathogenic bacteria, e.g., *Trueperella pyogenes* and *Escherichia coli*, invade the uterus, they trigger an inflammatory response [38,39,40], primarily driven by the immune system’s attempt to clear the infection, leading to release of pro-inflammatory cytokines and other mediators. Although this response is necessary to combat the infection, it also has adverse effects on reproductive health.

An important consequence of uterine inflammation is suppression of luteinizing hormone (LH) secretion from the anterior pituitary [42]. LH is crucial, as it promotes maturation of ovarian follicles and triggers ovulation. Inflammatory cytokines released during uterine infections interfere with LH release, disrupting the hormonal balance required for regular ovarian function [42,43,44]. Consequently, cows often have delays in resumption of estrous cycles and re-establishment of pregnancy.

Additionally, uterine inflammation can delay uterine involution. Although this should be complete within a few weeks postpartum, infection and inflammation cause delays. Incomplete uterine involution is less conducive to embryo implantation and survival, making it more difficult for cows to become and remain pregnant [45,46].

### 2.4. Cost of Uterine Disease

Uterine disease in dairy cows can be costly [47]. The cost of uterine disease (metritis with fetid, watery, red–brownish vaginal discharge ≤ 21 days after calving) ranged from $240 to $884 per case (including treatment, lost milk production, reproductive losses, and early culling [48]. For a typical 1000-cow herd with a 10% incidence of uterine infections, this (median, $398) could cost ~$40,000 per year. Further, the economic impact of purulent vaginal discharge, characterized as a mucopurulent, purulent, or reddish–brownish vaginal discharge at 28 ± 7 DIM, was estimated to be $202, range $152 to $265 [49] within a single lactation. Addressing and preventing uterine infections is essential for maintaining herd health and improving profitability and sustainability.

### 2.5. Timing of Intervention

An overview of clinical and economic considerations in the timing of postpartum uterine disease management, especially clinical and subclinical endometritis, is in Figure 1.

#### 2.5.1. Optimizing Uterine Recovery: A Veterinarian’s Approach

Managing uterine disease in dairy cows requires a careful balance of timing and intervention. Timing of the examination is critical to ensure that it occurs after uterine involution has had a chance to occur, but not so late that there is insufficient time for treatment and response before the start of the breeding period [50,51]. Typically, postpartum uterine involution takes ~5 wk [1]. However, it can be delayed or disrupted by infections, and the clinician must assess the cow at a point where treatment can effectively address any problems without interfering with this natural recovery process [52].

The goal of treating uterine disease is two-fold: first, to reverse the inflammatory changes within the uterus that impair fertility, and second, to enhance the cow’s ability to defend against further infections and support tissue repair. Inflammation within the uterus can disrupt normal hormonal signaling [53], delay uterine involution [52], and create an environment less conducive to embryo implantation [54,55]; therefore, reducing this inflammation is important. Clinicians typically use a combination of systemic and local therapies, e.g., antibiotics, anti-inflammatory drugs, or hormones, to reduce infection and inflammation, and to promote endometrial healing and restoration.

Treatment should aim to augment the immune response, enabling the uterus to resist infection and restore normal uterine function. This may involve improving general health through nutrition or addressing metabolic issues that could impair recovery. Additionally, uterine treatments should support repair of tissue damage caused by infection or retained fetal membranes, to ensure the uterus can support a pregnancy.

Ultimately, the clinician’s perspective is focused on maximizing fertility, minimizing long-term reproductive issues, and ensuring cows return to optimal health. Effective treatment improves the chances of conception and reduces financial losses associated with prolonged anestrus, breeding failures, and early culling. A well-timed and comprehensive approach to uterine disease can enhance herd productivity and profitability.

#### 2.5.2. Herd Management Perspective on Uterine Disease

From the perspective of dairy producers, effective management of uterine health is fundamentally linked to ensuring that cows conceive at a biologically optimal time and maintain economically sustainable calving intervals. Ideally, first breeding should occur within 60 to 90 days postpartum, when conception rates are highest and cows can return to a normal production cycle [52,53]. However, when uterine infections or other reproductive disorders are present, this window is often missed, resulting in prolonged calving intervals, decreased fertility, reduced herd productivity and increased culling.

The longer a cow is not pregnant after calving, the more costly it becomes due to lost milk production. Prolonged intervals also increase risk of early culling [56,57,58], reducing lifetime profitability and increasing replacement costs. A reduction in calving interval of 60 days increased daily milk production (1.51 and 1.11 kg/day) and during entire lactation, ~498 kg/lactation in high-production (12,500 kg in 305 days of lactation) and ~366 kg/lactation in moderate-production herds (9000 kg in 305 days of lactation) [59].

Producers want to breed cows as soon as they are biologically ready. Early detection and effective treatment of uterine diseases are key to breeding cows in a timely manner. Prompt and successful management of uterine infections shortens the interval from calving to conception, reducing the number of days the cow remains non-productive.

The ultimate objective is not simply achieving pregnancy but doing so in a manner that aligns both the biological potential for fertility and the economic goals of milk production and calf output [59]. Breeding cows at the optimal time minimizes reproductive delays, supports herd productivity, and maximizes profitability. Consequently, investment in preventive measures, timely diagnostic approaches, and effective therapeutic interventions for uterine diseases is essential. Such strategies not only enhance fertility outcomes but also reduce production costs and improve long-term profitability and sustainability of dairy operations.

## 3. Managing Uterine Disease

The problem with managing uterine disease in dairy cows lies in conflicting findings in the literature over the last five decades regarding effectiveness of diagnosis and treatment [14]. Research has produced a variety of conclusions, making it difficult for veterinarians and dairy producers to determine the most effective strategy. Some studies suggest that diagnosis has no benefit, leading to the idea that all animals should be treated proactively, irrespective of clinical signs of infection. This approach assumes that subclinical cases—those without obvious signs—might still be suffering from uterine disease, and treating all cows reduces long-term reproductive issues. Conversely, other research indicates that treatment for uterine disease has no benefit, implying that, in certain cases, treating the disease does not improve fertility or overall cow health [14]. This perspective questions the efficacy of interventions like antibiotics or hormone therapy, suggesting that the body’s natural recovery mechanisms might be sufficient. In contrast, in other studies, treatment for uterine disease does have benefit, particularly when the infection is diagnosed early and treated appropriately [13]. These studies emphasize the importance of intervention in improving fertility, reducing the risk of chronic health issues, and increasing milk production. Similarly, research indicating that diagnosis and treatment together have a benefit supports the idea that early detection of uterine infections, followed by targeted therapy, is the most effective approach. This lack of consensus creates confusion for dairy farm owners and veterinarians. Conflicting findings make it challenging to develop a standardized, evidence-based protocol for managing uterine disease, leaving producers to weigh risks and benefits of each approach, based on their specific circumstances [6].

### 3.1. Addressing Uterine Disease

#### 3.1.1. Diagnosis

The objectives of addressing uterine disease in dairy cows revolve around several key areas that need further clarity and improvement to optimize management and treatment strategies. A major issue is the lack of a standardized case definition for uterine disease, which encompasses a broad range of conditions, such as endometritis, metritis, and more sever uterine infections, with no universally accepted definition or diagnostic criteria. This makes it challenging for clinicians and producers to consistently identify affected cows and compare findings.

In general, accurate diagnosis of postpartum uterine diseases, such as metritis and endometritis, in dairy cows requires an integrated approach, combining clinical, physiological, histological, and hematological assessments [60,61]. Clinical signs are often the first indicators, with metritis typically presenting within 10 days postpartum as an enlarged, flaccid uterus, fetid red–brown discharge, and systemic illness including fever, anorexia, and reduced milk yield [62]. In contrast, endometritis, which occurs >21 days postpartum, lacks systemic signs but is characterized by mucopurulent or purulent vaginal discharge and delayed uterine involution [63]. Physiological alterations may include elevated rectal temperature (>39.5 °C), increased uterine size, and delayed cervical closure [60]. Histologically, endometritis is marked by infiltration of polymorphonuclear neutrophils (PMNs) into the endometrial epithelium and stroma, glandular atrophy, and epithelial disruption, whereas metritis may have extensive necrosis and microbial colonization [64,65]. Hematological changes can include leukocytosis or leukopenia, elevated acute-phase proteins (e.g., haptoglobin and serum amyloid A), and altered neutrophil-to-lymphocyte ratios, reflecting systemic inflammation [66,67]. For accurate diagnosis, proper sample collection is critical. Uterine cytology (e.g., cytobrush or low-volume lavage) should be performed under aseptic conditions to assess PMN percentages [64,68], whereas uterine swabs for culture or PCR must avoid contamination from the lower reproductive tract [39,40]. Blood samples should be collected via jugular venipuncture into EDTA and serum tubes for hematological and biochemical profiling. Integrating these diagnostic tools enhances the accuracy of disease identification, enabling timely and appropriate intervention.

Diagnostic methods are another critical area of focus [14]. Although there are various ways to assess uterine health—e.g., transrectal palpation, uterine swabs for microbial culture, cytology, or ultrasonography—their sensitivity, specificity, and practicality vary [13,68,69]. Determining the most accurate, cost-effective, and reliable diagnostic tools is essential for early detection and treatment of uterine disease. The challenge lies in selecting the appropriate method for diagnosing disease in a herd where clinical signs may not always be obvious, especially in subclinical cases.

Closely related to diagnostics are criteria to define uterine disease. Despite many definitions of uterine infection or disease, there is no consensus on what clinical signs or laboratory findings should trigger a diagnosis. Some criteria may include a purulent discharge, abnormal palpation findings, or evidence of systemic illness, whereas others focus on subclinical markers such as endometrial cytological evaluation, hormonal disruptions or delayed uterine involution. Defining specific criteria would allow for more standardized identification of uterine disease, improving the consistency of diagnosis and treatment protocols.

Regarding treatment, understanding both successes and failures is essential [14]. Studies on uterine disease treatments—antibiotics, anti-inflammatories, or hormones—have yielded mixed results [14]. Some treatments were beneficial in certain situations, whereas others have had limited efficacy or even potential harm, e.g., promoting antimicrobial resistance or disrupting the natural recovery process. Evaluating the success of various treatment protocols, as well as identifying why certain approaches fail, can help refine therapeutic strategies.

Lastly, exploring alternative methods for managing uterine disease is crucial. With concerns over antibiotic use and antimicrobial resistance, more attention is being given to alternative treatments, such as immunostimulants, probiotics, or non-antibiotic therapies, e.g., intrauterine infusion of plant extracts or natural compounds [70,71,72,73,74,75]. These may offer options for preventing or treating uterine infections with fewer side effects. Managing underlying metabolic conditions (e.g., ketosis and postparturient hypocalcemia) or improving herd health and nutrition may also reduce uterine disease [76,77].

The lack of a clear case definition for uterine disease in dairy cows is a challenge in both diagnosis and treatment. Currently, clinical or case definitions for uterine disease are often based on the presence or absence of vaginal discharge, commonly used as an indicator of uterine infection or inflammation. However, this criterion alone is not always sufficient to accurately diagnose uterine disease [74]. Vaginal discharge can vary widely in type, amount, and consistency, and although it may imply uterine infection, it does not necessarily reflect severity or a specific disease [78,79]. Additionally, some cows with uterine infections, especially if subclinical, may not have noticeable discharge, reducing diagnostic sensitivity.

Although “endometritis” and “metritis” are often used interchangeably, they refer to different conditions [80,81,82,83,84]. Endometritis, an inflammation of the endometrium, is typically a less severe form of uterine infection, whereas metritis involves inflammation of the entire uterine wall, often accompanied by systemic illness (e.g., fever and lethargy). However, the overlap in usage of these terms can cause confusion, making it difficult to distinguish between these conditions and to apply appropriate treatment strategies. Therefore, standardized definitions and clear diagnostic criteria, are critical.

Furthermore, diagnostic criteria are not always correlated with reproductive performance. For example, a cow with few or no clinical signs of uterine disease may have prolonged anestrus or delayed conception. Conversely, some cows with noticeable signs of uterine infection, e.g., discharge, may recover with limited impact on long-term reproductive health [85]. Therefore, diagnosing uterine disease based on clinical signs or discharge does not consistently predict current or future reproductive status. Consequently, there is a pressing need for a more robust case definition that goes beyond clinical signs, incorporating factors such as the severity of uterine inflammation, microbiological findings, hormonal disruptions, and their direct correlation with reproductive outcomes. Establishing a clearer definition would not only improve the accuracy of diagnosis but also provide a basis for evaluating effectiveness of treatment and management strategies, improving herd health and more consistent fertility outcomes.

A standardized classification system for uterine diseases could improve diagnosis, treatment, and research efforts [86]. One classification includes conditions like puerperal metritis, a severe inflammation of the uterus that occurs shortly after calving, often leading to systemic illness. Metritis, a broader term, refers to inflammation of the uterine lining, of varying degrees of severity. Clinical endometritis, which involves inflammation of the endometrium and is often associated with visible clinical signs like vaginal discharge, is another key condition. Additionally, subclinical endometritis, without obvious clinical signs, can still negatively affect reproductive performance and may be harder to diagnose. Finally, pyometra, is characterized by purulent material within the uterus. Establishing clear definitions and classifications of these uterine diseases would not only streamline clinical practice but also facilitate more targeted research aimed at improving animal health and reproductive outcomes.

Case definitions for uterine diseases, particularly endometritis and metritis, have remained largely unchanged. Currently, the clinical or case definition is primarily based on the presence or absence of vaginal discharge; however, that can be limiting in accurately diagnosing endometritis and metritis. This approach fails to capture cases that may not have clinical signs but still have underlying reproductive issues. Furthermore, “endometritis” and “metritis” are often used interchangeably, which can lead to confusion and inconsistency in diagnosis. Another area of variability is the definition of subclinical endometritis, where the percentage of polymorphonuclear neutrophils (PMNs) used as a cutoff to diagnose the condition varies from 6 to 18% [68,87]. This inconsistency in defining subclinical endometritis can complicate research and clinical management, highlighting the need for standardized, more precise criteria to improve diagnosis and treatment outcomes for cattle with uterine diseases.

Diagnostic methods and criteria for bovine uterine diseases are varied, with a huge range in sensitivity and specificity. A primary indicator is an abnormal fetid discharge, which is often associated with conditions like metritis. In addition, rectal temperature is an important diagnostic tool, as fever can signal systemic infection or inflammation, common in metritis. Transrectal uterine palpation helps identify abnormalities such as an enlarged or inflamed uterus, which can indicate infection or other uterine conditions; however, diagnostic accuracy varies with clinician experience. Vaginoscopy is also used to detect abnormal discharge or structural changes in the reproductive tract [80,81]. Uterine culture provides information about microbial causes of uterine diseases [60,82,83,84,85]. Similarly, uterine biopsy can offer insights into tissue health and chronic inflammation or infection [85]. Endometrial cytology, i.e., collecting and examining cells from the uterine lining, is crucial for diagnosing endometritis and differentiating between clinical and subclinical disease [68,86]. Finally, ultrasonography is a non-invasive and detailed assessment of fluid accumulation, endometrial thickness, uterine abnormalities [13,68], or pyometra. Together, these diagnostic methods provide a comprehensive approach to evaluating uterine health, though integration of multiple tools is often necessary for accurate diagnosis and to guide treatment.

To diagnose uterine diseases such as endometritis, various diagnostic techniques have been evaluated for their sensitivity (Sn) and specificity (Sp), with median days open (MDO, interval from calving to conception) serving as a gold standard. Palpation of the cervix, particularly when it exceeds 7.5 cm, had 17% sensitivity (Sn) and 82% specificity (Sp) when performed at 28 days postpartum, with a MDO of 120 days in milk (DIM) [63]. Vaginoscopy, which focuses on detecting a purulent discharge, had higher specificity (94 to 87%) but lower sensitivity (21 to 7%), with MDOs ranging from 120 to 150 DIM [13,87,88]. Endometrial biopsy had a sensitivity of 92% and a specificity of 77%, with an MDO of 128 DIM [65,89]; however, it caused infertility [65]. Endometrial cytology, with lavage or cytobrush, is also commonly used. A lavage or a cytobrush with >8% PMN had specificity of 84 to 93% but limited sensitivity (14 to 25%) depending on the method [13,86]. Ultrasonography is another critical tool for uterine disease diagnosis. Fluid > 3 mm in the uterine lumen had high specificity (93%) but low sensitivity (10%) with MDO 150 days [13] whereas endometrial thickness >7 mm has high specificity (90%) but poor sensitivity (11%) with MDO 150 days [13]. These diagnostic techniques, when combined, help veterinarians accurately diagnose and manage uterine diseases, though the choice of method may vary based on clinical circumstances and the desired outcome [90].

For diagnosing subclinical endometritis, several diagnostic methods have been evaluated, with endometrial cytology and ultrasonography being key tools. Endometrial cytology, where the percentage of polymorphonuclear neutrophils (PMNs) > 18% is used as a diagnostic criterion, had a sensitivity of 36% and a specificity of 94% at a mean day of onset (MDO) of 132 days [68]. This method is highly specific but has lower sensitivity, meaning it is very accurate when the condition is present but may miss some cases of subclinical endometritis (Table 1). Conversely, ultrasonography, which assesses the presence of fluid in the uterine lumen, had a more balanced sensitivity and specificity, both ~56% at MDO 132 days [68]. Although ultrasonography is less specific than cytology, it provides valuable information about the physical state of the uterus and can be used to identify fluid accumulation (often associated with subclinical endometritis). Combining these diagnostic tools can offer a more comprehensive approach to diagnosing subclinical endometritis, though it is clear that cytology remains a more specific test for confirming the condition.

#### 3.1.2. Treatment

Treatment of metritis has been widely examined. Intrauterine (IU) treatments with antibiotics like oxytetracycline or ampicillin do not have any significant benefit over systemic antibiotics such as ceftiofur or penicillin [62,91,92,93]. This suggests that systemic antibiotics are just as effective, if not more so, in managing the condition. Additionally, in high-risk cows during the first 3 days in milk (DIM), giving ceftiofur when fever is detected reduced incidence of subsequent fetid discharge, a hallmark of metritis [94,95,96,97].

In general, treatment of postpartum uterine diseases in dairy cows, including endometritis and metritis, typically involves either local (intrauterine) or systemic administration of therapeutics, each with specific advantages and limitations. Local treatments, such as intrauterine infusion of antibiotics (e.g., cephapirin), target the uterus directly, allowing high local drug concentrations with minimal systemic exposure [60,86]. However, their efficacy can be reduced by the presence of purulent material, altered uterine pH, and poor diffusion into deeper uterine layers, limiting action against pathogens residing in the endometrial stroma [61,98]. Systemic treatments, commonly involving antibiotics like ceftiofur or anti-inflammatory drugs, offer broader distribution and are especially useful in cases with systemic signs of illness (e.g., fever and anorexia) [99,100]. Nevertheless, systemic administration may lead to lower drug concentrations in the uterus and poses a higher risk of antimicrobial resistance and drug residues in milk [101].

On a molecular level, effective treatment has been associated with modulation of inflammatory pathways, including downregulation of pro-inflammatory cytokines (e.g., IL-1β and TNF-α) and restoration of normal endometrial gene expression patterns involved in tissue repair and immune resolution [6,102]. However, incomplete resolution of inflammation at the molecular level—even in clinically recovered cows—suggests that current treatments may not fully restore uterine homeostasis, highlighting the need for more targeted and integrative therapeutic strategies [66].

Fever resolution was achieved in 67–77% of treated cows within 5 to 10 days after treatment, although resolution of the fetid odor associated with metritis was more challenging [98]. Therefore, although fever can be effectively managed with appropriate antibiotic therapy, the fetid discharge may require more time or different therapeutic approaches. Moreover, ceftiofur HCl, 2.2 mg/kg IM every 24 h for 5 days, significantly reduced the incidence of fever and fetid discharge in cows with retained fetal membranes, starting as early as 1 DIM, even when no fever was present [94]. However, this treatment did not affect time to pregnancy, suggesting that although ceftiofur can improve health outcomes in terms of fever and discharge, it does not necessarily accelerate reproductive recovery [103]. Systemic ceftiofur treatment, especially when started early, is an effective option for managing metritis and its associated symptoms [103,104].

Treatment for clinical and subclinical endometritis often involves PGF2α and intrauterine antibiotics, but the specific treatment can depend on the presence or absence of a corpus luteum (CL) [105,106,107]. It is generally believed that cows with endometritis who have an existing CL should be treated with PGF2α, as this induces luteolysis [105,106,107], decreasing progesterone, which can suppress uterine contractility and immune response, promoting healing and improving fertility [108].

Conversely, cows with endometritis but without a CL (therefore lacking elevated progesterone that would suppress immune status) may benefit more from an intrauterine infusion of antibiotics, e.g., ceftiofur or cephapirin [93,94,95,96,97]. In such cases, delivering the antibiotic directly to the site of infection increases drug concentrations in the affected area, improving chances of clearing the bacterial infection and promoting uterine health.

In cows classified as high risk for uterine disease, Ceftiofur crystalline free acid did not prevent the onset of metritis [109], implying that prophylactic treatment with Ceftiofur may not be beneficial in reducing the incidence of uterine diseases. Furthermore, in that study, administration of PGF2α at 5 and 7 weeks postpartum did not mitigate negative effects of cytological endometritis (CYTO, ≥6% PMN) or postpartum vaginal discharge (PVD) on reproductive performance. This was relevant, because PGF2α is commonly used to address uterine inflammation and promote cyclicity, but in this case, it failed to overcome reproductive challenges.

These findings underlined the complexity of uterine disease management and the importance of a tailored approach. For cows at high risk or those with CYTO or PVD, Ceftiofur and PGF2α may not always improve uterine health or reproductive performance.

#### 3.1.3. Treatment Failure

There are several reasons why treatments for uterine diseases in cows, e.g., endometritis and metritis, are not always effective. These factors can be related to antibiotic limitations, bacterial characteristics, and regulatory constraints. Here are some key challenges:

##### Minimal Inhibitory Concentrations (MICs)

The effectiveness of antibiotics in treating bovine metritis and endometritis depends on a minimum inhibitory concentration (MIC) at the site of infection that inhibits or kills the causative pathogens [110]. For instance, Sheldon et al. (2004) demonstrated that *Escherichia coli*, a common pathogen in cases of acute metritis, requires sustained concentrations of ceftiofur above the MIC to achieve clinical cure [111]. However, intrauterine conditions such as purulent discharge, altered pH, and inflammatory exudate can impede antibiotic penetration and activity [112]. Drillich et al. (2006) reported that systemic administration of cephalosporins was less effective when the uterine environment had a high volume of exudate, decreasing intrauterine antibiotic concentration [113]. Additionally, chronic infections involving *Trueperella pyogenes* may require higher or prolonged antibiotic exposure due to bacterial persistence and biofilm formation [114,115]. If the antibiotic does not achieve or sustain effective concentrations in uterine tissue, particularly in the face of these challenges, treatment is likely to fail.

##### Choice of Antibiotics and Regulations

The choice of antibiotics used in treating uterine infections is often influenced by regulations regarding antibiotic use in food-producing animals [116,117]. Many regions have strict guidelines and withdrawal periods to prevent antibiotic residues in milk and meat, which can limit antibiotics available for uterine diseases [112,116,117,118]. Furthermore, available antibiotics may not always be the most effective against pathogens causing uterine infection, leading to suboptimal treatment outcomes [119].

##### Bacterial Antibiotic Resistance

One of the biggest challenges is antibiotic resistance, where bacteria acquire mechanisms to survive and thrive despite antibiotics [116,120]. Overuse or misuse of antibiotics can promote development of resistant strains, making it increasingly difficult to treat infections. In cases of metritis and endometritis, resistant bacteria can survive initial rounds of treatment, leading to recurrent or chronic infections that are harder to resolve [121]. Resistant pathogens may not respond to standard antibiotics typically used in uterine infections, necessitating more targeted or potent treatments [14,121].

##### Biofilm Formation

Another critical factor contributing to persistence and treatment resistance of uterine infections is the ability of certain bacteria to form biofilms—structured communities of microorganisms that adhere to surfaces and are embedded within a self-produced matrix composed of polysaccharides, proteins, and extracellular DNA [122,123]. This biofilm matrix is a physical and biochemical barrier, protecting bacteria from environmental threats, including antibiotics and host immune responses. As a result, bacteria residing within biofilms can be up to 1000 times more resistant to antimicrobial agents compared to their free-floating, or planktonic, counterparts.

In the context of postpartum uterine diseases, biofilms can form on the endometrial lining, creating a persistent source of infection that is particularly difficult to eradicate [124,125]. This structural protection limits antibiotic penetration and enables bacteria to survive treatment, contributing to chronic or recurrent infections. Additionally, bacteria within biofilms undergo genotypic and phenotypic adaptations, including altered gene expression that enhances virulence, suppresses immune detection, and increases resistance to conventional therapies. These adaptations not only complicate clinical management, but may also hinder accurate detection using standard diagnostic techniques, as biofilm-associated bacteria are often distinctly different from their planktonic forms.

##### Transition from Planktonic to Biofilm State

Bacteria in the uterus often start in a planktonic state (free-floating) but transition to a biofilm state when they encounter surfaces like endometrium [126]. This transition is accompanied by profound changes in behavior and genotypic characteristics, making them less susceptible to conventional antibiotic treatment [16,122,123,124,125,126]. Biofilm-associated bacteria are more difficult to target with antibiotics because they are shielded by the protective biofilm matrix, and the bacteria within the biofilm often have lower metabolic rates, making them less responsive to antibiotics that target actively dividing cells [127].

##### Sequestration and Support of Bacteria

The biofilm matrix is a barrier that sequesters bacteria and shields them from antibiotics and the host immune response [128]. This makes bacteria more resilient, even in the presence of antimicrobials. The biofilm not only protects bacteria from direct effects of antibiotics but also promotes persistence of infections by a slow release of bacteria from the biofilm, potentially causing recurrent infections despite an apparent cure [129].

MICs, antibiotic resistance, biofilm formation, and bacterial adaptation, create challenges for treating uterine infections in cows. To improve outcomes, there may need to be better diagnostics to identify the specific pathogens involved, development of treatments targeting biofilm-associated bacteria, and antibiotic stewardship to minimize resistance. Understanding biological processes that enable bacteria to survive and persist in the uterus could lead to more effective treatments and better reproductive performance.

The MIC is the lowest antibiotic concentration to inhibit growth of a particular pathogen. MIC values for common antibiotics against common uterine pathogens are in Table 2.

Selected cephalosporins, such as ceftiofur and cephapirin, have lower MIC values and thus have potential effectiveness in treating uterine bacterial infections in cattle, as they are effective against a wide range of pathogens, including both Gram-positive and Gram-negative bacteria commonly implicated in endometritis and metritis. However, development of antibiotic resistance, particularly among Gram-negative strict anaerobes (e.g., *Fusobacterium necrophorum* and *Bacteroides* spp.), poses a challenge. Resistance mechanisms, such as beta-lactamase production, efflux pumps, and porin channel mutations, can reduce effectiveness of cephalosporins, despite low MICs. This resistance is especially concerning when biofilm formation or mixed infections are present, as these further complicate treatment. Empirical use of cephalosporins may not always be successful. Therefore, it is crucial to perform susceptibility testing (e.g., MIC testing) to identify the most effective antibiotics, especially in persistent or severe infections. Combination therapy may be necessary in cases of resistance, pairing cephalosporins with other antibiotics such as fluoroquinolones or tetracyclines. Alternative treatments [70,72,73,74,75], including non-antibiotic adjuncts like PGF2α for luteolysis or uterine lavage, can enhance antibiotic efficacy, and intrauterine infusions of targeted therapies may require higher doses or longer durations in resistant cases. Additionally, a preventive approach, focusing on managing the transition period, improving nutrition, and minimizing stress, can reduce the risk of uterine infections and lessen reliance on antibiotics [130].

In a study [124] to detect multi-drug resistance (MDR) genes in *Trueperella pyogenes* (*16S rRNA*, *intI 1*, *intI 2*, and cassette genes) and biofilm virulence factor (VF) genes in *Escherichia coli* (*Agn43aCFT073* and *Agn43bCFT073*) isolated from the uterus of postpartum dairy cows, 12 out of 40 cows (30%) tested positive for either MDR genes in *Trueperella pyogenes* or biofilm VF genes in *Escherichia coli*, suggesting that these factors may contribute to poor responses to antibiotic treatments. The presence of MDR genes in *Trueperella pyogenes* indicates that this pathogen is resistant to multiple antibiotics, limiting effectiveness of common treatments for uterine infections. Likewise, detection of biofilm VF genes in *Escherichia coli* implies biofilm formation. Bacterial resistance and biofilm formation are important barriers to successful treatment of postpartum uterine diseases in dairy cows, as conventional antibiotic therapy is often ineffective. These findings highlight the need for alternative approaches, such as more targeted therapies based on susceptibility testing, the use of biofilm-disrupting agents, and adjunctive treatments that enhance the immune response. In addition, preventive strategies, including better management of the transition period and vaccination against biofilm-forming pathogens, could reduce uterine infections and improve reproductive health in dairy herds.

Several promising approaches are being explored to address challenges posed by antibiotic resistance, biofilm formation, etc. For example, PGF2α, has shown promise in some cows with uterine disease, as it induces luteolysis and improves uterine health by clearing out infection and reducing inflammation. In addition, mucolytic drugs, e.g., N-acetylcysteine, are being investigated for uterine irrigation [131], as they can break down biofilm matrices in the uterus, making bacterial pathogens more susceptible to antibiotics and the immune system, improving treatment outcomes [131,132]. Another novel approach involves intrauterine infusion of a hypertonic solution (50% dextrose) [133], that improved reproductive outcomes, as indicated by better pregnancy/AI rates (29.8% in cows treated with dextrose, compared to 21.1% in controls, and lower rates in cows with clinical endometritis). Furthermore, supplementation with omega-3 fatty acids, vitamin E and selenium is being considered, as these nutrients have anti-inflammatory and antioxidant properties that may support uterine recovery and improve overall health [134,135]. In vitro studies on nano-drug carriers could deliver targeted antimicrobial treatments directly to biofilm-associated bacteria, providing a more effective solution to persistent uterine infections [136]. These exciting advancements offer new hope for improving treatment of uterine diseases, reducing reliance on antibiotics, and enhancing reproductive health in dairy cattle.

Body condition score (BCS) significantly influences postpartum uterine health; cows with lower BCS have a higher incidences of uterine disease and persistent inflammation due to altered metabolic and cytokine profiles [137]. A reduced BCS is associated with both pro- and anti-inflammatory cytokine increases, creating a feedback loop that prolongs inflammation and complicates recovery [137]. Mucin 1 (*MUC1*), an innate immune effector, has a critical role in embryo attachment; its overexpression has been linked to infertility in humans and cows [54,138,139]. In cows, excessive *MUC1* expression—due to inflammation or poor embryo quality, can impede implantation, highlighting mucin regulation as a key factor in reproductive success [54,138,139].

## 4. Role of Proteomics, Metabolomics, and miRNAs in Uterine Diseases in Postpartum Dairy Cows

Systems biology approaches, e.g., proteomics, metabolomics, and microRNA (miRNA) profiling, have opened new avenues for understanding pathophysiology of uterine diseases and for developing novel diagnostic and therapeutic strategies (Figure 2).

Integration of omics technologies—encompassing genomics, transcriptomics, proteomics, metabolomics, and microbiomics—has begun to redefine understanding of uterine diseases such as metritis and endometritis in dairy cows. At the level of pathogenesis, transcriptomic analyses have revealed differential expression of genes related to inflammation, immune regulation, and tissue remodeling in the endometrium of affected cows. For example, IL1B, TNF, and CXCL8 are consistently upregulated in endometrial tissue during clinical endometritis, indicating a persistent pro-inflammatory state that disrupts uterine involution and healing [12,60]. Proteomic profiling complements these findings by identifying altered abundance of proteins involved in neutrophil activation, extracellular matrix degradation, and oxidative stress, offering mechanistic insights into tissue damage and repair processes [140,141,142]. Metabolomic studies also identified systemic metabolic alterations—e.g., elevated β-hydroxybutyrate and reduced glucose—that predispose cows to impaired immune responses and increase susceptibility to uterine infection [141,142]. Microbiomics have revolutionized understanding of the uterine microbial ecosystem; cows developing metritis or endometritis often have distinct microbial communities dominated by *Fusobacterium necrophorum*, Bacteroides, and *Trueperella pyogenes*, compared to healthy cows with more diverse and stable uterine microbiota [8]. These omics-derived insights are clarifying disease mechanisms and enabling development of biomarker-based diagnostics and individualized therapeutic strategies, e.g., targeted antimicrobials, immunomodulators, or microbiome-based interventions.

### 4.1. Proteomics: Understanding Functional Protein Changes

Proteomics facilitates comprehensive analysis of protein expression, modification, and interactions in uterine tissues and fluids, during health and disease. In postpartum cows, the uterus undergoes rapid changes, and disturbances in protein profiles can be indicative of inflammatory responses, bacterial invasion, and tissue remodeling. Advances in proteomics have revealed critical insights into how uterine diseases alter protein expression in the uterus, affecting local immune responses and the reproductive environment.

Proteomic analyses of uterine tissues, luminal fluid, and exosomes confirmed significant differences in protein profiles between healthy and diseased cows. For example, cows with uterine disease had increased inflammatory proteins, such as acute-phase proteins and matrix metalloproteinases, which contribute to tissue damage and impair uterine repair [140]. Based on pathway analysis, infected exosomes had more proteins involved in structural molecule activity and immune system processes than non-infected exosomal protein. Additionally, proteins from infected exosomes were involved in unique pathways, e.g., angiogenesis and integrin signaling pathway [141]. Chronic inflammation disrupted endometrial receptivity, reducing successful implantation [143,144,145].

Based on uterine luminal fluid proteomes, disease states are associated with higher abundance of proteins involved in oxidative stress, metabolic dysregulation, and protein degradation pathways [146]. Such changes may compromise the uterine microenvironment, reducing embryo viability and increasing early embryonic loss.

Proteomics has revealed associations between fertility potential and protein expression. Cows with genetically low fertility had increased levels of proteins related to folate metabolism (e.g., folate receptor alpha) and vitamin A transport (e.g., retinol-binding protein), markers of disrupted metabolic function in the uterus [147]. Further, immune system-related proteins—lactotransferrin and chromogranin A—were enriched in low-fertility cows together with interferon tau-3 and interferon tau-2. These disruptions, compounded by uterine infection, can exacerbate subfertility.

### 4.2. Metabolomics: Capturing Biochemical Changes

Metabolomics investigates metabolic profiles of biological fluids (e.g., blood or uterine lochia). In postpartum uterine diseases, the metabolic fingerprint reflects changes in host–pathogen interactions, immune activation, and tissue damage [148,149]. Recent advances in metabolomics have provided critical insights into how these disorders disrupt reproductive physiology and fertility. These diseases trigger systemic and localized metabolic shifts that adversely affect the uterine environment, embryo development, and reproductive success [148,150].

Metabolomic profiling of blood, uterine fluid, and vaginal mucus in cows with uterine infections revealed altered key metabolites, including amino acids, lipids, and energy-related compounds. Metabolomic analysis of vaginal discharge identified > 400 metabolites, with key differences between healthy and metritis cows, as early as 6 h postpartum [151]. Distinct metabolite patterns at 4 and 7 DIM revealed dysregulation in glutathione, amino acid and energy metabolism [151]. Additionally, cows cured of metritis had elevated lignoceric, maleic, and malic acids [151]. These findings highlighted early metabolic biomarkers linked to metritis onset, progression, and treatment response.

For instance, cows diagnosed with uterine disease had increased concentrations of inflammatory mediators and decreased levels of essential amino acids (e.g., arginine and glutamine) in uterine secretions [152,153]. These indicate impaired immune regulation and tissue repair, both essential for restoring a receptive uterine environment.

Additionally, metabolomics documented that uterine diseases disrupt energy metabolism, as evidenced by altered glucose, pyruvate, and short-chain fatty acid concentrations in affected cows [153,154]. These can reduce availability of energy substrates necessary for early embryo development, perhaps increasing embryonic mortality.

Metabolic signatures have also been linked to fertility status. High-fertility cows typically maintain more stable metabolic profiles postpartum, whereas low-fertility cows have greater deviations in metabolites related to oxidative stress, steroid metabolism, and inflammation [148]. Uterine disease often exacerbates these deviations, leading to further declines in reproductive performance.

### 4.3. Role of MicroRNAs in Postpartum Uterine Disease in Dairy Cows

Serum profiling of 84 bovine-specific miRNAs revealed that 30 miRNAs were dysregulated (*p* ≤ 0.05; fold change ≥ 2) in cows with metritis compared to healthy controls [21]. Of these, 16 miRNAs were upregulated, including bta-miR-15b, miR-17-3p, miR-148a, and miR-101, whereas 14, such as bta-miR-148b, miR-122, and miR-200b, were downregulated. The dysregulation likely reflects host responses to microbial infection and inflammation. Notably, bta-miR-15b, linked to Gram-negative bacterial endotoxins like lipopolysaccharide (LPS), was highly upregulated and may inhibit genes involved in uterine involution. Target gene prediction revealed that these miRNAs regulate hundreds of genes involved in apoptosis, immune responses, tissue remodeling, and inflammation. For example, bta-miR-148a and bta-miR-101 potentially disrupt matrix degradation and immune signaling. Downregulation of bta-miR-200b, which regulates TGF-β signaling, suggests a link to impaired uterine remodeling. Both miR-122 and miR-199a, associated with liver function and pregnancy maintenance, respectively, were also significantly altered, underscoring broader systemic effects. Functional classification of target genes using PANTHER, indicated involvement in key biological processes: immune response, inflammation, apoptosis, and tissue repair—vital for resolving uterine infection and facilitating recovery. Interestingly, miRNAs from the same family (e.g., miR-148a vs. miR-148b) exhibited divergent expression and functions.

Another study [155] investigated whether subclinical endometritis in dairy cows can impair fertility by disrupting the uterine environment, potentially through altered regulation of miRNAs. This study investigated whether subclinical endometritis affects uterine miRNA expression and their associated biological pathways. Using a genome-wide miRNA PCR array (354 primers), researchers compared miRNA profiles in uterine cytobrush samples from cows with and without subclinical endometritis. Twenty-three miRNAs were differentially expressed in affected cows. An in vitro endometrial cell culture model challenged with lipopolysaccharide (LPS) had similar miRNA expression patterns, supporting in vivo findings. Based on Ingenuity Pathway Analysis, these miRNAs regulated key pathways involved in inflammation, cell proliferation, movement, the cell cycle, and apoptosis—critical processes in endometrial health and disease. Expression analysis confirmed these target genes were in the endometrium, and luciferase reporter assays validated several predicted miRNA–gene interactions.

Manipulating miRNA expression is a novel therapeutic strategy. Using miRNA mimics or antagomiRs (anti-miRNA oligonucleotides) to restore normal gene regulation is being explored, including in cattle [155]. However, challenges such as efficient delivery to target tissues, specificity, and avoidance of off-target effects remain as barriers to clinical application.

## 5. Uterine Disease and Impacts on Early Embryo Development

Uterine diseases are well known for causing reduced fertility and extended calving intervals. Furthermore, accumulating evidence suggests a direct detrimental effect on early embryo development and survival, even after clinical resolution of the disease (Figure 3).

The bovine uterus is a dynamic environment supporting early embryo development from fertilization through blastocyst formation and implantation. During this window (~days 0–16 d post-insemination), the embryo is highly sensitive to changes in: uterine secretions (histotroph), immune modulation, and endometrial receptivity [156,157,158]. Uterine disease disrupts this delicate balance by altering cytokine expression, uterine pH, microbial load, and the metabolic profile of uterine fluids, compromising embryo development, elongation, and maternal recognition of pregnancy [14,24].

Induced endometrial inflammation alters uterine health, conceptus development, and molecular signaling associated with early pregnancy in lactating cows and heifers. Induced endometrial inflammation increased subclinical endometritis, elevated PMN percentages, and disrupted immune homeostasis without affecting ovulation or pregnancy rates by day 16. However, conceptuses from cows with induced endometrial inflammation were shorter and secreted less IFNT, suggesting compromised development. Furthermore, endometrial inflammation impaired expression of genes related to growth (*IGF1*, *GHR*), metabolism (*LDHA*, *G6PD*), and antioxidant defense (*SOD1*, *GPX1*), while increasing genes associated with immune activation (*TLR9*, *MYD88*) and stress (*WEE1*, *FOS*). This indicated a less favorable uterine environment, with widespread reductions in histotroph metabolites, particularly lipids and amino acids critical for conceptus nourishment and signaling (Figure 3).

In heifers, induced endometrial inflammation similarly increased inflammatory markers and reduced expression of *IFNT*, *MTOR*, and nutrient transport genes in the conceptus, reinforcing that inflammation disrupts developmental competence even with embryo transfer. In summary, uterine inflammation impaired conceptus elongation and maternal recognition of pregnancy through transcriptomic and metabolic disruptions. Therefore, good uterine health during the peri-conceptional period is critical to promote reproductive success in dairy cattle.

Kasimanickam et al. [159] investigated gene expression profiles in the endometrium and Day 16 conceptuses of repeat breeder dairy cows, with or without subclinical endometritis (SCE), after embryo transfer. Cows with SCE had altered expression of several implantation-related genes, including interferon-tau (*IFNT*), ISGs (*ISG15*, *RSAD2*, *CTSL1*), and glucose transporter *SLC2A1*, suggesting impaired maternal-conceptus communication [24]. Furthermore, poor-quality embryos transferred into SCE-affected cows reduced conceptus development, as evidenced by altered morphometry and gene profiles. Therefore, both embryo quality and uterine health affected reproductive success [160].

## 6. Nutritional Strategies to Improve Reproduction and Combat Uterine Infections in Modern Dairy Cows

Modern dairy cows are genetically selected for high milk yields, particularly in early lactation, often resulting in negative energy balance (NEB) and excessive mobilization of body reserves. Cows can produce an average of 324 kg of milk (up to 550 kg) from body reserves [161]. This energy deficit during the first 5–7 weeks postpartum contributes to metabolic stress, disease susceptibility, and lower reproductive performance [161].

A major reproductive consequence of metabolic stress is increased risk of uterine infections that delay ovarian cyclicity and reduce conception rates [53,161,162]. These infections often arise from impaired immune function during the transition period [163]. Therefore, nutritional strategies must address energy balance and support the cow’s immune response to reduce infection risk and enhance overall fertility.

### 6.1. Transition Cow Management and Immune Support

A foundational nutritional strategy is optimizing transition cow management to support immune function (Figure 4). This includes minimizing dietary and social stressors, promoting consistent dry matter intake, and ensuring appropriate micronutrient supplementation [164]. Key elements involve maintaining an appropriate body condition score before calving and avoiding drastic decreases postpartum, as poor energy status impairs neutrophil function—critical for clearing postpartum uterine contamination [165].

Vitamins and trace minerals are crucial for immune defense [166]. Supplementing with vitamin E and selenium reduced the incidence of retained fetal membranes and uterine infections, both linked to decreased fertility [167]. Injecting 1000 IU of vitamin E weekly during the last 3 weeks of pregnancy significantly reduced pregnancy loss and postpartum complications [167]. Vitamin E is a potent antioxidant that enhances neutrophil and macrophage function, whereas selenium supports glutathione peroxidase, a key enzyme protecting immune cells from oxidative damage.

### 6.2. Dietary Energy and Fertility

Glucogenic diets immediately postpartum increases insulin, promoting resumption of ovarian activity [168,169]. However, these should be halted before breeding to avoid negative effects on oocyte quality and progesterone metabolism [170,171]. Properly timed energy support enhances metabolic resilience and may reduce uterine infections by preventing excessive fat mobilization and associated immunosuppression [164].

High-fat feeding aimed at reducing NEB by increasing dietary energy density have had inconsistent results. Omega-6 fatty acids may promote uterine involution by prostaglandin F2α stimulation, but omega-3s may reduce inflammatory damage and support embryo survival [135]. However, saturated fatty acids from mobilized body fat in early lactation were linked to impaired oocyte and embryo quality, and potentially weakened immune function [172].

### 6.3. Protein and Uterine Health

Excess dietary protein or imbalanced amino acid supply, particularly causing high blood urea nitrogen (BUN), has been associated with reduced uterine environment quality and fertility [171]. Elevated urea may impair uterine pH and embryo development [151]. Soy-based protein sources contain isoflavones that may alter ovarian and luteal function. Recent findings suggest negative effects on steroidogenesis and angiogenesis in the corpus luteum, although progesterone concentrations were not significantly affected [173].

### 6.4. Energy Balance and Uterine Immunity

Negative energy balance (NEB) disrupts immune balance and slows postpartum uterine recovery through heightened inflammatory signaling [174]. NEB enhances uterine inflammation by upregulating *TLR4* signaling, increasing cytokines, chemokines, and neutrophil recruitment. In cows under severe NEB, genes linked to interferons and antiviral defense (e.g., *MX1* and *GBP1*) were elevated. *MMPs* involved in tissue remodeling were also significantly upregulated, suggesting delayed uterine repair. Increased *IL-1* and *AHSG* expression further stimulated *MMP* activity.

### 6.5. Micronutrients and Uterine Immunity

Beyond vitamin E and selenium, other micronutrients such as zinc, copper, and manganese have essential roles in uterine repair and immune function [175]. Zinc supports epithelial barrier integrity and wound healing, whereas copper is involved in neutrophil activity. Deficiencies in these elements, although harder to detect, may predispose cows to persistent infections and suboptimal reproductive outcomes [176].

## 7. Conclusions

In conclusion, the likelihood of failing to identify cows with uterine disease through a single examination is high, as cows that appear clinically normal early in the postpartum period may subsequently develop uterine infections. This highlights the importance of repeated postpartum monitoring. The increasing challenge of antimicrobial resistance, coupled with bacterial biofilm formation, further emphasizes the urgent need for alternative and more sustainable treatment strategies. From a herd management perspective, excellent transition nutrition, an optimal body condition score at calving, and supporting cows through the early postpartum period are key practical measures for reducing the incidence of uterine disease and enhancing reproductive performance.

In terms of future directions, integration of omics technologies—particularly multi-omics approaches combining proteomics, metabolomics, and miRNA profiling—offers powerful tools for unraveling the complex pathogenesis of uterine disorders. These approaches will not only enhance our mechanistic understanding but also facilitate development of precision diagnostics and individualized therapeutic interventions. Moving forward, practical implementation of omics-informed tools in herd-level health programs could revolutionize disease prediction, early intervention, and treatment outcomes. Moreover, insights from bovine models may also have translational relevance to human postpartum uterine health, offering broader biomedical implications. Ultimately, bridging fundamental research with applied clinical practice will be essential to improving both animal health and the economic sustainability of dairy operations.

## Figures and Tables

**Figure 1 animals-15-02841-f001:**
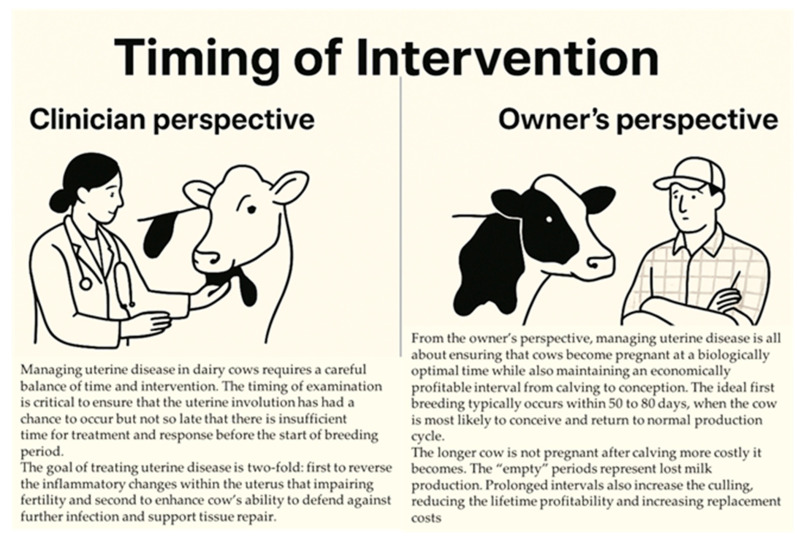
Clinical and Economic Considerations in the Timing of Postpartum Uterine Disease Management.

**Figure 2 animals-15-02841-f002:**
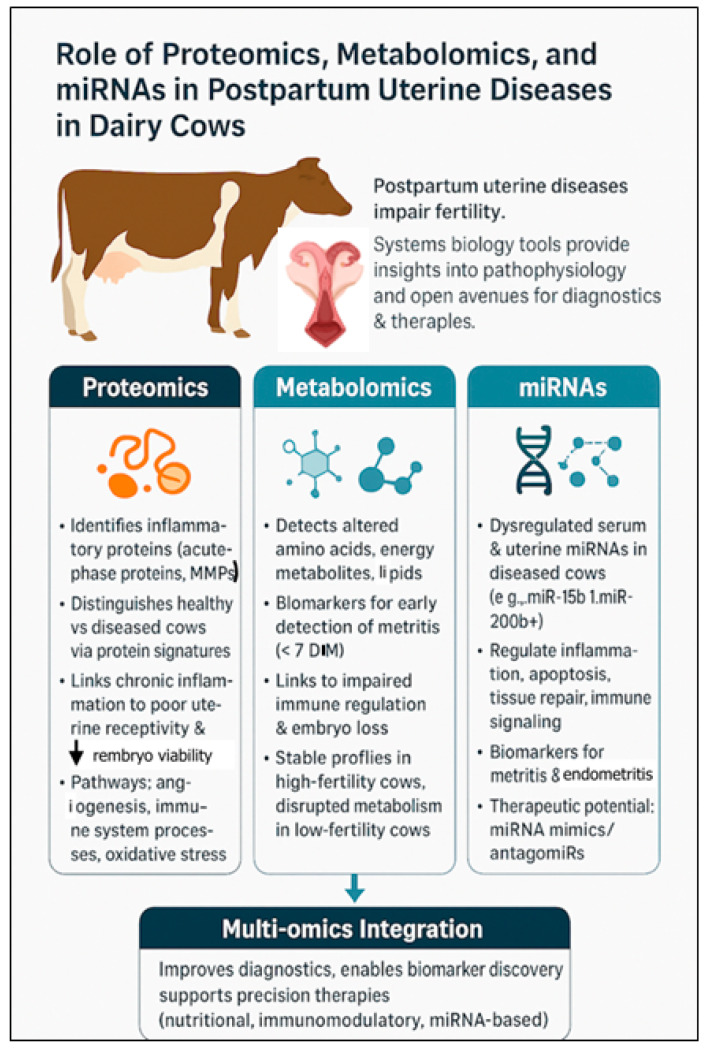
Multi-Omics Insights into Postpartum Uterine Diseases in Dairy Cows. ↓, decrease.

**Figure 3 animals-15-02841-f003:**
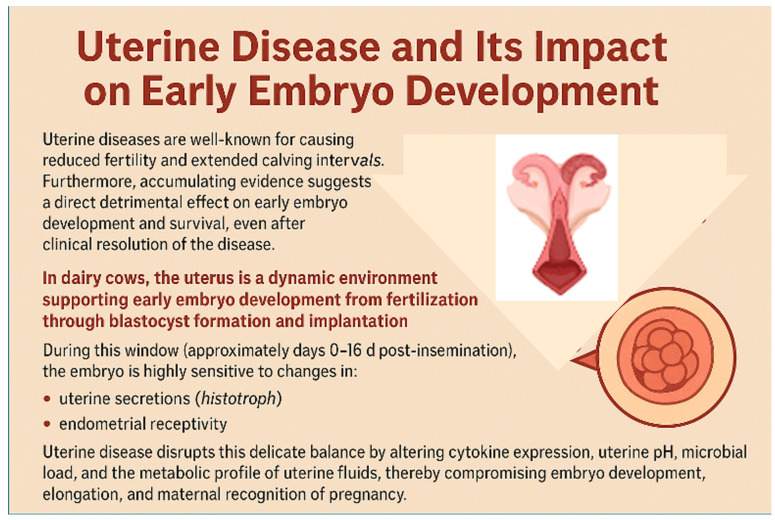
Disruption of Early Embryo Development by Uterine Disease: Mechanisms and Fertility Implications.

**Figure 4 animals-15-02841-f004:**
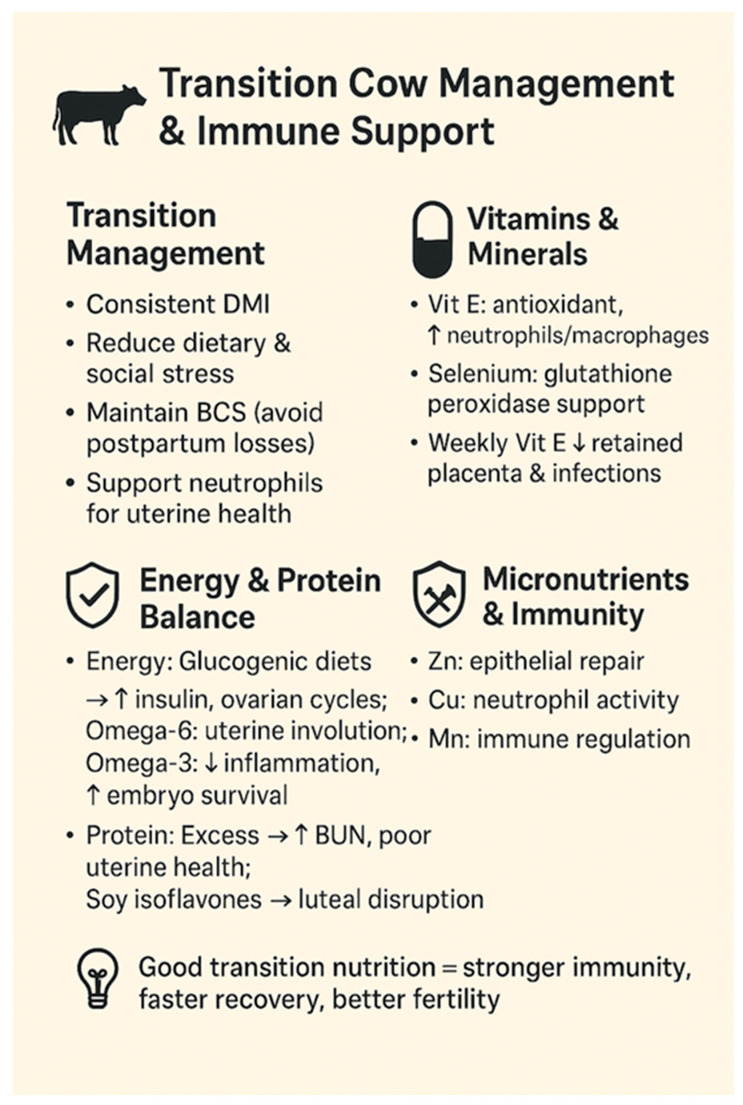
Optimizing transition cow nutrition to enhance immune resilience and reproductive success. ↓—decrease; ↑—increase.

**Table 1 animals-15-02841-t001:** Uterine disease classification, diagnostic technique, sensitivity and specificity.

Uterine Disease	Diagnostic Technique	Median Days Open	Sensitivity	Specificity	Reference
Endometritis	Palpation (>7.5 cm cervix)	120	17% (Low)	82% (High)	[63]
	Vaginoscopy (Purulent discharge)	120	21% (Low)	91% (High)	[63]
		150	7% (Low)	89% (High)	[13]
	Biopsy	128	92% (High)	77% (Medium)	[65]
	Endometrial cytology (Lavage %PMN > 8%)	150	14% (Low)	84% (High)	[13]
	Endometrial cytology (Cytobrush %PMN > 8%)	150	13% (Low)	91% (High)	[13,86]
	Endometrial cytology (Cytobrush %PMN > 6%)	150	25% (Low)	86% (High)	[86]
	Ultrasonography (Uterine lumen fluid > 3 mm)	150	10% (Low)	93% (High)	[13]
	Ultrasonography (Endometrial thickness > 7 mm)	150	1% (Low)	90% (High)	[13]
Subclinical Endometritis	Endometrial cytology (%PMN > 18%)	132	36% (Low)	94% (High)	[68]
	Ultrasonography (Uterine lumen fluid)	132	56% (Medium)	56% (Medium)	[68]

**Table 2 animals-15-02841-t002:** Minimum Inhibitory Concentration (MIC) values for some antibiotics against common pathogens of the bovine uterus.

Pathogen	Antibiotic	MIC_50_ (µg/mL)	MIC_90_ (µg/mL)
*Escherichia coli*	Cefquinome	<0.06 *	<0.06
	Enrofloxacin	<0.06	<0.06
	Oxytetracycline	1	>32
	Cephapirin	4	8
	Ceftiofur	0.5	0.5
*Trueperella pyogenes*	Cefquinome	<0.06	0.125
	Enrofloxacin	1	1
	Oxytetracycline	16	32
	Cephapirin	<0.06	<0.06
	Ceftiofur	<0.06	0.125
*Fusobacterium necrophorum*	Cefquinome	0.5	>32
	Enrofloxacin	8	8
	Oxytetracycline	1	16
	Cephapirin	<0.06	2
	Ceftiofur	<0.06	0.125

* Bacteria tested were inhibited by a concentration of antibiotic < 0.06 µg/mL.

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
