# Peer review of "From Infection to Infertility: Diagnostic, Therapeutic, and Molecular Perspectives on Postpartum Metritis and Endometritis in Dairy Cows"

_animals, 2025, doi:10.3390/ani15192841_

Round 1
Reviewer 1 Report
Comments and Suggestions for Authors
The review titled “From Infection to Infertility: Diagnosis, Treatment, and Molecular Insights into Postpartum Uterine Disease in Dairy Cows” (animals-3875961) synthesizes current knowledge on proteomics, metabolomics, and miRNAs in postpartum uterine diseases and highlights future directions for research and clinical applications. It is a paper with a unique focus that makes a significant contribution to the field. However, the following comments need to be considered by the authors:
In the keywords section on line 47, five keywords are repeated from the title. It is suggested that the author replace them with others or delete them, because the title and keywords appear simultaneously in the retrieval system. There is no need to waste retrieval resources. On the contrary, introducing more keywords can expand the influence of the paper.
Is the picture in the Graphical Abstract original or sourced from elsewhere? I noticed that there are still words or graphics at the bottom left edge of the picture.
The content of Biofilm Formation on line 487 can be expanded, because quorum sensing can regulate biofilm formation. Can the treatment involved in this paper be elaborated and resolved from the perspective of quorum sensing? The current summary is too brief, and the latest research progress extracted is insufficient, especially the research reports from the past five years.
Why is 0.06 chosen as the threshold for MIC50 in Table 2 on line 523?
Good Luck!
Author Response
The review titled “From Infection to Infertility: Diagnosis, Treatment, and Molecular Insights into Postpartum Uterine Disease in Dairy Cows” (animals-3875961) synthesizes current knowledge on proteomics, metabolomics, and miRNAs in postpartum uterine diseases and highlights future directions for research and clinical applications. It is a paper with a unique focus that makes a significant contribution to the field.
Author: We sincerely appreciate the reviewer’s encouraging and positive feedback on our manuscript. We are pleased that the reviewer recognizes the unique focus and contribution of our review to the field. We have carefully addressed the reviewer’s comments on a point-by-point basis in the revised manuscript.
In the keywords section on line 47, five keywords are repeated from the title. It is suggested that the author replace them with others or delete them, because the title and keywords appear simultaneously in the retrieval system. There is no need to waste retrieval resources. On the contrary, introducing more keywords can expand the influence of the paper.
Authors: The keywords has been changed as per suggestions
Is the picture in the Graphical Abstract original or sourced from elsewhere? I noticed that there are still words or graphics at the bottom left edge of the picture.
Authors: The graphical abstract has been modified.
The content of Biofilm Formation on line 487 can be expanded, because quorum sensing can regulate biofilm formation. Can the treatment involved in this paper be elaborated and resolved from the perspective of quorum sensing? The current summary is too brief, and the latest research progress extracted is insufficient, especially the research reports from the past five years.
Authors: The biofilm section is expanded as suggested
Why is 0.06 chosen as the threshold for MIC50 in Table 2 on line 523?
Authors: Thanks for the clarification. We did not choose. 0.06 µg/mL means that 0.06 micrograms of a specific antimicrobial per milliliter is the lowest concentration required to prevent visible growth of a particular microorganism in a laboratory setting.
Reviewer 2 Report
Comments and Suggestions for Authors
The review article titled “From Infection to Infertility: Diagnosis, Treatment, and Molecular Insights into Postpartum Uterine Disease in Dairy Cows” aims to address key aspects related to uterine infections, diagnostic methods, and treatment strategies in dairy cows. In general, the article is overly broad and contains information extending beyond the stated scope of the title. The major concern is that the review attempts to cover multiple issues in a single framework, which reduces clarity and coherence, especially when proper citations are lacking. Several sections require refinement, contextual adjustments, and improved alignment with the objectives. Major revisions are recommended. Detailed section-wise comments are as follows:
Title: The title appears acceptable but requires modification. The term uterine disease is too broad, as it encompasses a variety of conditions beyond metritis or endometritis. A more specific title focusing on the conditions actually discussed would be clearer and more attractive.
Simple Summary and Abstract: Both sections mirror each other. The Simple Summary should be written in a style accessible to a broader audience, rather than using language suited for a scientific readership.
Graphical Abstract: This should be revised by incorporating an image of the bovine reproductive system instead of the human reproductive system. Pathogens should be depicted in the same frame, and diagnostic/treatment tools separated visually. The inclusion of mRNA is disproportionate, as its discussion occupies only a very small portion of the manuscript.
Introduction:
The first paragraph should emphasize the economic losses associated with uterine infections. The discussion of pathophysiological aspects is redundant.
This section should concisely highlight: (i) economic impact, (ii) involved conditions, (iii) risk factors/etiology, (iv) diagnostic approaches, (v) treatment options, and (vi) the rationale for writing this review.
Lines 96–103: These sentences are irrelevant in this context and should be deleted.
Subheading 2. Postpartum Uterine Disease:
The major issue is that the authors simultaneously discuss septic metritis and endometritis, making it difficult to follow the objectives. These conditions should be classified and discussed separately under the same general heading.
2.2. Elaborate how E. coli establishes infection, facilitates metritis, and progresses to endometritis, including the associated molecular changes.
2.3. The subheading Casual Mechanism should be changed to Pathophysiology.
2.5.1 and 2.5.2. These subsections should be simplified and made more concise. Although the expression differs, the content is repetitive (lines 221–226 vs. 238–244).
The titles of 2.5.1 and 2.5.2 use the term perspective, which is too vague. The word targets would be more appropriate.
Subheading 3. Diagnosis:
Lines 248–267 lack references and must be supported by citations.
3.1.1. This section should include details on physical signs, physiological alterations, histological findings, and hematological changes in clinical samples. Additionally, guidance on proper sample collection should be added for accurate diagnosis.
3.1.2. Treatment: Treatment should be categorized into local vs. systemic protocols for managing endometritis and metritis, including the limitations of each route of administration. Information on molecular changes in response to treatments should also be provided.
Lines 464–471: These statements must be supported with specific examples and references.
Treatment Failure Factors: This section currently contains general statements unsupported by citations. Substantial improvement is needed.
Subheadings 4.1 and 4.2 (Omics):
Introductory paragraphs on general aspects of proteomics and metabolomics are unnecessary. Instead, omics-related changes should be integrated within the discussion of pathogenesis, diagnosis, and treatment, rather than being presented in a separate standalone section.
Subheadings 5 and 6: These sections are outside the defined scope of the review and should be revised or removed.
Conclusion:
The conclusion summarizes the key features of the review adequately but could be strengthened by highlighting practical implications and future perspectives.
Author Response
We sincerely thank the reviewer for their thoughtful and constructive comments on our manuscript. We appreciate the reviewer’s time and expertise in evaluating our work. Below, we have addressed each point raised, providing clarifications and outlining the revisions made in response to the suggestions.
Title: The title appears acceptable but requires modification. The term uterine disease is too broad, as it encompasses a variety of conditions beyond metritis or endometritis. A more specific title focusing on the conditions actually discussed would be clearer and more attractive.
Authors: Thank you for the suggestion. We have revised the title to better reflect the specific conditions discussed in the review.
Revised Title: From Infection to Infertility: Diagnostic, Therapeutic, and Molecular Perspectives on Postpartum Metritis and Endometritis in Dairy Cows
Simple Summary and Abstract: Both sections mirror each other. The Simple Summary should be written in a style accessible to a broader audience, rather than using language suited for a scientific readership.
Authors: We appreciate this observation. The Simple Summary has been revised using more accessible language to better suit a general audience, while maintaining its informative value.
Graphical Abstract: This should be revised by incorporating an image of the bovine reproductive system instead of the human reproductive system. Pathogens should be depicted in the same frame, and diagnostic/treatment tools separated visually. The inclusion of mRNA is disproportionate, as its discussion occupies only a very small portion of the manuscript.
Authors: Thank you for this valuable suggestion. We have redesigned the graphical abstract to include the suggestions.
Introduction:
The first paragraph should emphasize the economic losses associated with uterine infections. The discussion of pathophysiological aspects is redundant. This section should concisely highlight: (i) economic impact, (ii) involved conditions, (iii) risk factors/etiology, (iv) diagnostic approaches, (v) treatment options, and (vi) the rationale for writing this review.
Authors: With respect to the request for a concise overview of economic impact, conditions, risk factors, diagnostic approaches, treatment options, and rationale, we respectfully note that each of these aspects is already covered in detail under separate headings and subheadings in the main text of the review. In order to avoid unnecessary redundancy, we have chosen to maintain the current Introduction structure, which briefly outlines the significance of PUD and then transitions into the molecular focus of the review. In addition, we have revised the rationale for the review.
Lines 96–103: These sentences are irrelevant in this context and should be deleted.
Authors: Regarding lines 96–103, we respectfully retained these sentences because they address the impact of postpartum uterine diseases (PUD) on early embryo development, which is highly relevant to the overall scope of this review. Since reproductive performance and fertility outcomes are directly affected by uterine health, highlighting the consequences of uterine inflammation on endometrial receptivity, oocyte quality, and embryo survival provides essential context. This connection underscores the broader significance of PUD beyond immediate clinical symptoms, aligning directly with the review’s goal of integrating molecular insights with reproductive outcomes. We believe that maintaining these sentences strengthens the Introduction by bridging the clinical and economic aspects of uterine disease with its ultimate consequence—reduced fertility—which is of central concern to dairy herd productivity.
Subheading 2. Postpartum Uterine Disease: The major issue is that the authors simultaneously discuss septic metritis and endometritis, making it difficult to follow the objectives. These conditions should be classified and discussed separately under the same general heading.
Authors: Please refer Section 3.1.1. In Section 3.1.1, our intention was to highlight the continuum and overlap between the terminology.
2.2. Elaborate how E. coli establishes infection, facilitates metritis, and progresses to endometritis, including the associated molecular changes.
We thank the reviewer for this comment. However, Section 2.2 of the manuscript is specifically dedicated to discussing risk factors for postpartum uterine diseases. A detailed description of how E. coli establishes infection and progresses to metritis or endometritis—including molecular mechanisms—would shift the focus away from risk factors and introduce material that is more appropriately addressed in pathogen-specific reviews. We deliberately structured Section 2.2 to remain concise and focused on risk determinants rather than detailed pathogen biology. We respectfully believe that expanding on E. coli pathogenesis in this section would be beyond scope and create redundancy with existing literature. We hope the reviewer will agree that keeping Section 2.2 focused on risk factors maintains clarity and strengthens the overall organization of the manuscript.
2.3. The subheading Casual Mechanism should be changed to Pathophysiology.
Authors: The subtitle was changed to "Pathophysiological mechanisms".
2.5.1 and 2.5.2. These subsections should be simplified and made more concise. Although the expression differs, the content is repetitive (lines 221–226 vs. 238–244).
Authors: We thank the reviewer for this observation. Upon careful review, we respectfully note that these two subsections are not redundant but rather complementary in scope. Specifically: Lines 221–226 (Subsection 2.5.1) focus on the biological timeline of reproduction, emphasizing the optimal postpartum breeding window and the consequences of delayed conception. Lines 238–244 (Subsection 2.5.2) highlight the economic rationale and management strategies, addressing how delayed breeding impacts profitability, productivity, and sustainability. To improve clarity and ensure these distinctions are more apparent to readers, we have rephrased both subsections for conciseness and to reduce any perceived overlap. The revisions make the unique focus of each subsection clearer, while preserving the important link between biological and economic perspectives. We hope the reviewer will find the revised version more streamlined and better aligned with the manuscript’s objectives.
The titles of 2.5.1 and 2.5.2 use the term perspective, which is too vague. The word targets would be more appropriate.
Authors: Focused titles were included as suggested.
Subheading 3. Diagnosis:
Lines 248–267 lack references and must be supported by citations.
Authors: References included.
3.1.1. This section should include details on physical signs, physiological alterations, histological findings, and hematological changes in clinical samples. Additionally, guidance on proper sample collection should be added for accurate diagnosis.
Authors: A paragraph has been include focusing these criteria
3.1.2. Treatment: Treatment should be categorized into local vs. systemic protocols for managing endometritis and metritis, including the limitations of each route of administration. Information on molecular changes in response to treatments should also be provided.
Authors: A paragraph has been include focusing these criteria
Lines 464–471: These statements must be supported with specific examples and references.
Authors: This paragraph has been modified to incorporate the reviewer suggestion.
Treatment Failure Factors: This section currently contains general statements unsupported by citations. Substantial improvement is needed.
Authors: Thank you for your comment. We would like to clarify that the section in question introduces six sub-sections, each discussing a specific treatment failure factor. Within those sub-sections, we have provided supporting references and examples to substantiate the points made.
Subheadings 4.1 and 4.2 (Omics):
Introductory paragraphs on general aspects of proteomics and metabolomics are unnecessary. Instead, omics-related changes should be integrated within the discussion of pathogenesis, diagnosis, and treatment, rather than being presented in a separate standalone section.
Subheadings 5 and 6: These sections are outside the defined scope of the review and should be revised or removed.
Authors: We appreciate the reviewer’s observation. However, we respectfully maintain that the sections in question are directly relevant to the overall scope of the review, as they address the impact of uterine disease on fertility—a core outcome of clinical and biological importance in dairy production systems. Understanding how uterine disease influences reproductive performance is essential for contextualizing its pathophysiology, diagnosis, and treatment. Therefore, we believe this content contributes meaningfully to the objectives of the review and have retained it in the revised manuscript.
Conclusion: The conclusion summarizes the key features of the review adequately but could be strengthened by highlighting practical implications and future perspectives.
Authors: The conclusion was modified to incorporate the reviewer suggestion.
Round 2
Reviewer 2 Report
Comments and Suggestions for Authors
Although the authors have revised the manuscript and adequately addressed most of the raised queries, some concerns remain unanswered. In its current form, the manuscript meets the required standards and is recommended for publication after incorporating the suggested revisions.
Subheadings 4.1 and 4.2 (Omics):
The introductory paragraphs on general aspects of proteomics and metabolomics are unnecessary. Instead, omics-related changes should be integrated into the discussion of pathogenesis, diagnosis, and treatment, rather than being presented in a separate standalone section.
Subheadings 5 and 6:
These sections fall outside the defined scope of the review and should either be revised substantially or removed.
Figures:
Figures 2 and 3 should be revised, and a tract figure from cows should be added for better illustration.
Author Response
Subheadings 4.1 and 4.2 (Omics):
The introductory paragraphs on general aspects of proteomics and metabolomics are unnecessary. Instead, omics-related changes should be integrated into the discussion of pathogenesis, diagnosis, and treatment, rather than being presented in a separate standalone section.
Authors: We appreciate these comments. As our paper has a strong clinical focus, we expect that many of the readers will be clinicians with limited knowledge of Omics. These sections include a brief theoretical background and the examples are directly relevant to the main topic. Therefore, these sections have two functions: provide background knowledge to ensure that everyone has the background and context and then provide specific information relevant to the topic, which should be of broad interest. Therefore, we feel that these sections have inherent value and we have chosen to retain them.
Subheadings 5 and 6:
These sections fall outside the defined scope of the review and should either be revised substantially or removed.
Authors: Thank you for your comments.
First, here are the section titles:
- Uterine Disease and Impacts on Early Embryo Development
- Nutritional Strategies to Improve Reproduction and Combat Uterine Infections in Modern Dairy Cows
Section 5 describes the consequences of uterine disease, specifically how it reduces reproductive performance. We feel that this is of relevance and have cited some data regarding cellular/subcellular aspects.
Section 6 addresses how nutrition and management can be used to reduce the risk and impacts of uterine disease. Historically, clinical management of uterine disease was focussed on detection and treatment, typically with antibiotics or antiseptics. Although this approach remains relevant, in this paper, we have outlined the limitations of both diagnostics and conventional treatments. Furthermore, in the current context of increasing antimicrobial resistance and increasing demands to decrease the use of antimicrobials, we feel that a more holistic approach, including nutrition and management, is appropriate and relevant.
Finally, we are confident that the contents of Sections 5 and 6 are consistent with our title.
Figures:
Figures 2 and 3 should be revised, and a tract figure from cows should be added for better illustration.
Authors: We agree and have made these modifications.